# Adaptive Fast-and-Slow Visual Program Reasoning for Long-Form VideoQA

## Abstract

Large language models (LLMs) have shown promise in generating program workflows for visual tasks. However, previous approaches often rely on closed-source models, lack systematic reasoning, and struggle with long-form video question answering (videoQA). To address these challenges, we introduce the FS-VisPR framework, an adaptive visual program reasoning approach that balances fast reasoning for simple queries with slow reasoning for difficult ones. First, we design efficient visual modules (e.g., key clip retrieval and subtitle retrieval) to support long-form video tasks. Then, we construct a diverse and high-quality fast-slow reasoning dataset with a strong LLM to align open-source language models' ability to generate visual program workflows as FS-LLM. Next, we design a fast-slow reasoning framework with FS-LLM: Simple queries are directly solved by VideoLLMs, while difficult ones invoke visual program reasoning, motivated by human-like reasoning processes. During this process, low-confidence fast-thinking answers will trigger a second-stage slow-reasoning process, and a fallback mechanism to fast reasoning is activated if the program execution fails. Moreover, we improve visual programs through parameter search during both training and inference. By adjusting the parameters of the visual modules within the program, multiple variants are generated: during training, programs that yield correct answers are selected, while during inference, the program with the highest confidence result is applied. Experiments show that FS-VisPR improves both efficiency and reliability in visual program workflows. It achieves 50.4% accuracy on LVBench, surpassing GPT-4o, matching the performance of Qwen2.5VL-72B on VideoMME.

## 1 Introduction

Video Question Answering (VideoQA) requires models to reason over dynamic visual content to answer natural language queries (Yu et al., 2019; Ning et al., 2023; Chen et al., 2023; Fang et al., 2024; Li et al., 2024c; Fu et al., 2024; Li et al., 2024b). Recent advances in Video Large Language Models (VideoLLMs) have shown impressive progress in this area (Li et al., 2023; Zhang et al., 2023b; Lin et al., 2023; Li et al., 2024a; Bai et al., 2025b; Zhang et al., 2024c). However, these models still struggle with long-form videos (Wu et al., 2024; Fu et al., 2024; Wang et al., 2024a), where query-relevant information is sparse and widely distributed across the video. Processing hundreds or thousands of frames requires a high computational cost, and the lack of task decomposition reduces both planning and interpretability. A promising direction is to leverage LLMs to generate visual program workflows that integrate powerful vision modules (Gupta & Kembhavi, 2023; Subramanian et al., 2023; Surís et al., 2023; Mahmood et al., 2024; Choudhury et al., 2023). By executing structured modules, this approach enables step-by-step reasoning and provides interpretability. However, prior efforts have focused mainly on short clips or image-based tasks (Choudhury et al., 2023; Surís et al., 2023) and lack an efficient module design tailored for long-form VideoQA. Furthermore, reliance on closed-source models with few-shot prompting (Brown et al., 2020) and the absence of adaptive reasoning strategies hinder both efficiency and scalability. Intuitively, not all questions require visual program reasoning: For simple queries, a VideoLLM can often provide reliable answers directly (Cheng et al., 2023; Zhu et al., 2023). Inspired by dual-process theories of human reasoning (Evans, 2008; Evans & Stanovich, 2013; Xiong et al., 2023; Taubenfeld et al., 2025), we observe that the confidence of a VideoLLM response can serve as an effective signal

| Dataset | Cor. | InCor. | Δ |
|---|---|---|---|
| LongVideoBench | 0.74 | 0.45 | 0.29 |
| VideoMME | 0.72 | 0.48 | 0.24 |
| LVBench | 0.67 | 0.46 | 0.22 |

Table 1: Average confidence scores for correct and incorrect predictions, including the gap (Correct − Incorrect). Confidence is derived from the decoded probabilities of options.

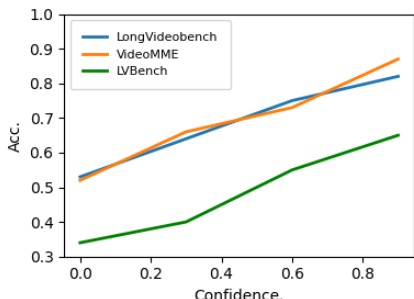

Figure 1: Prediction Accuracy for Samples Above the Confidence Threshold.

of response reliability (see Figure 1 and Table 1). This motivates an adaptive fast-and-slow reasoning paradigm, where simple queries are handled via fast reasoning, while difficult cases invoke program-based reasoning.

Building on these insights, we propose FS-VisPR, an adaptive visual program reasoning framework for long-form videoQA. We first design a set of efficient long-video modules, including key clip retrieval, subtitle-audio retrieval, object detection, trimming, and cropping etc., to enable programmatic reasoning over both temporal and multimodal cues. Next, we employ strong LLMs to generate quality and diverse visual program workflows (which can reach the correct choices and the module diversity), including the planning annotations and module calls. We identify simple queries that VideoLLM can answer directly and integrate fast-reasoning logic code into the visual program workflows. This logic enables VideoLLM to provide immediate responses with confidence scores, which are used to determine whether to return the answer directly. Then, we fine-tune the open-source language model as FS-LLM, aligning it with both fast and slow reasoning abilities. During inference, FS-LLM adopts robust strategies: low-confidence fast answers trigger a second-stage slow reasoning process, and failures in slow reasoning are back to fast reasoning. Moreover, inspired by human hyperparameter search to optimize programs, we introduce a parameter search for the modules to further enhance robustness. By varying the parameter values (e.g., `Top_k` = {1, 3, 5}), multiple (three) candidate programs are generated. During training, programs that yield correct answers are retained, while during inference, the candidate program with the highest confidence result is applied. Our main contributions can be summarized as follows:

- We design effective vision modules for long-form VideoQA, enabling efficient frame and subtitle retrieval, and construct diverse, high-quality visual workflows with strong LLMs to align open-source models with the ability to fast-slow reasoning as FS-LLM.

- We propose FS-VisPR, a fast-slow reasoning framework that leverages response confidence as a control signal. FS-LLM generates the visual program workflows, where VideoLLM directly handles simple queries as fast reasoning, while difficult queries are addressed using visual program reasoning, achieving adaptive reasoning across varying difficulty levels.

- We develop a module parameter search mechanism for visual program adjustment, generating diverse program variants during training to reach the correct answer, and selecting the most confident result at inference.

Extensive experiments on long-form VideoQA benchmarks show that FS-VisPR is both effective and efficient. It achieves 50.4% accuracy on LVBench, surpassing GPT-4o, and outperforms Qwen2.5VL-72B by about 2% on LongVideoBench, all while relying on a 7B VideoLLM.

## 2 RELATED WORK

### 2.1 LARGE LANGUAGE MODELS AND VISUAL PROGRAM REASONING

Large Language Models (LLMs) have made significant progress in language understanding and reasoning (Huang et al., 2022; Wang et al., 2022; Wei et al., 2022; Kojima et al., 2022). Beyond text, LLMs are increasingly applied to program generation. Early works focused on mapping natural

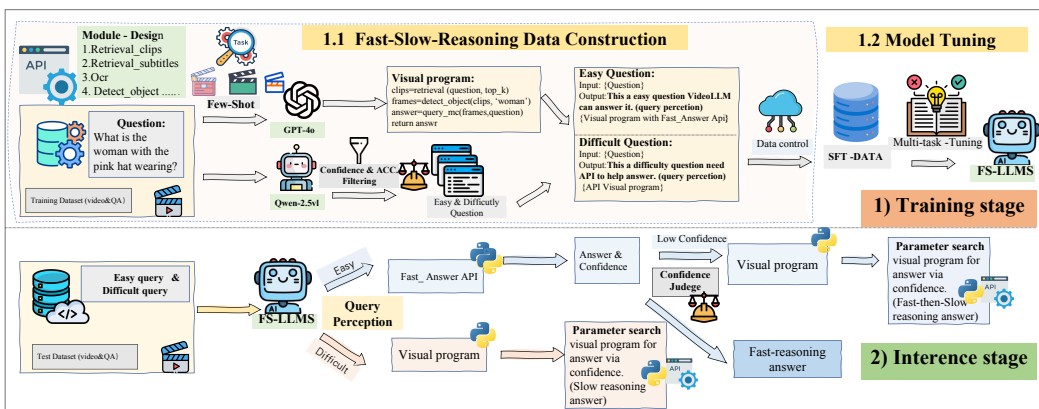

Figure 2: Fast–Slow Visual Program Reasoning framework: Fast–slow dataset construction aligns model as FS-LLMs to perceive query difficulty and adopt dual-reasoning strategies during inference.

language prompts to code (Chen et al., 2021; Li et al., 2022), while later studies extended this to programmatic workflows for multi-step tasks (Gao et al., 2023; Chen et al., 2022), across domains such as math (Chen et al., 2022), planning (Silver et al., 2022; 2024), and multimodal reasoning (Johnson et al., 2017; Gupta & Kembhavi, 2023; Surís et al., 2023; Choudhury et al., 2023). For instance, ViperGPT (Surís et al., 2023) integrates visual modules for image and short-video QA (Choudhury et al., 2023), while VURF (Mahmood et al., 2024) enhances program reliability. These efforts position LLMs as general-purpose planners for decomposing complex tasks into interpretable steps. However, many rely on closed-source models and resource-intensive prompting, often lacking effective visual modules for long-form VideoQA, which limits scalability.

## 2.2 LONG-FORM VIDEOQA AND VIDEO-LLMS

Long-form VideoQA requires reasoning over extended sequences and capturing temporal and causal dependencies (Xiao et al., 2021; Wu et al., 2024; Fu et al., 2024; Wang et al., 2024a). Recent Video-LLMs extend temporal visual encoders for joint spatial-temporal reasoning (Bai et al., 2025b; Zhang et al., 2024a;b; Li et al., 2024a; Cheng et al., 2024), but face memory and computational bottlenecks with long videos. To address this, some strategies use captioning or keyframe summarization to create textual representations for LLMs (Zhang et al., 2023a; Wang et al., 2024b; 2025), which improve scalability but can lose fine-grained temporal details and require multiple inference steps. These limitations motivate FS-VisPR, which dynamically focuses on key segments and employs visual program reasoning.

## 2.3 DUAL-PROCESS REASONING IN AI MODELS

Dual-process theory distinguishes between fast (intuitive) and slow (deliberative) reasoning (Evans, 2003; 2008). Recent AI research has adopted fast-slow paradigms (Xiao et al., 2025; Sun et al., 2025; Zhang et al., 2025; Sun et al., 2024), where fast reasoning efficiently handles simple queries, and slow reasoning is used for complex tasks. Previous methods often treat these modes separately and do not integrate them with visual program generation, limiting efficiency-accuracy trade-offs. Our framework adaptively switches between fast and slow reasoning based on model confidence: queries are initially addressed by fast reasoning, triggering slow reasoning when confidence is low, with a fallback to fast reasoning if the slow reasoning fails.

## 3 METHOD

In this section, we present FS-VisPR, an adaptive visual-program reasoning framework for long-form VideoQA. The central idea is simple and effective: the FS-LLM first estimates the query's difficulty based on its learned perception. For queries deemed difficult, FS-VisPR directly employs slow reasoning, utilizing structured visual programs with external modules. For queries judged as

easy, the model attempts fast reasoning by directly generating an answer together with a confidence score. If confidence is high, the fast answer is returned; otherwise, FS-VisPR falls back to slow reasoning for more reliable computation. Additionally, if the visual program cannot be executed, the fast reasoning answer is returned as a fallback.

## 3.1 CONFIDENCE ANALYSIS

Given a video $V = \{f_1, \ldots, f_T\}$ and a query $Q$, the VideoLLM $p_\theta$ autoregressively generates an answer sequence $\hat{A} = [\hat{a}_1, \ldots, \hat{a}_m]$. We define the model confidence as the exponential of the average log-likelihood of the decoded tokens:

$$\text{Conf}(V, Q) = \exp\left(\frac{1}{m} \sum_{t=1}^{m} \log p_\theta(\hat{a}_t \mid V, Q, \hat{a}_{<t})\right) \tag{1}$$

To examine the reliability of this measure, we conduct experiments in three long-form VideoQA benchmarks using QwenVL-2.5 as the backbone. As shown in Table 1, the model exhibits substantially higher confidence for correct predictions (mean $\sim 0.70$) than for incorrect ones (mean $\sim 0.45$), with a gap of about 25%. Furthermore, Figure 1 shows that accuracy increases monotonically with the confidence threshold, confirming that confidence is a strong indicator of answer quality and motivating FS-VisPR.

| Modules | Parameters | Description |
|---|---|---|
| GetClips | video_path, query, top_k | Retrieves the top-$k$ video clips most relevant to the given query. |
| GetSubtitles | video_path, query, top_k | Retrieves the top-$k$ subtitle segments most relevant to the given query. |
| TrimBefore | video_path, timestamp, intervals | Retrieves frames preceding the specified timestamp, with duration defined by *intervals*. |
| TrimAfter | video_path, timestamp, intervals | Retrieves frames following the specified timestamp, with duration defined by *intervals*. |
| TrimRange | video_path, start, end | Retrieves frames within the temporal range from *start* to *end*. |
| QueryMC | frames, query, choices | Answers a multiple-choice question using the given frames and candidate choices. |
| QueryYN | frames, query | Answers a binary (yes/no) question using visual evidence in the frames. |
| RunOCR | frame | Performs optical character recognition on the input frame and returns recognized text. |
| DetectObject | frame, text | Detects objects in the frame conditioned on a textual prompt; returns bounding boxes. |
| GetSubsRange | video_path, start, end | Retrieves subtitles within the temporal range from *start* to *end*. |
| GetCapsRange | video_path, start, end | Retrieves captions within the temporal range from *start* to *end*. |
| GetSubtitleHint | video_path, query | Retrieves subtitle segments or hints semantically relevant to the query. |
| Crop | frame, box | Crops the specified region from a frame to enable focused analysis. |
| ExtractFrames | video_path | Extracts all frames from the video for subsequent processing. |
| SplitVideo | video_path | Segments the video into candidate intervals based on scene structure. |
| FastThink | video_path, query | VideoLLM directly generate the answer and confidence score. |

Table 2: Modules and their parameters in FS-VisPR for long-form VideoQA.

## 3.2 LONG-VIDEO MODULE DESIGN

To enable modular reasoning over long-form videos, FS-VisPR builds upon structured and efficient modules $\mathcal{M}$ and an associated parameter space $\mathcal{Q}$ (Table 2). Each module $m \in \mathcal{M}$ is designed as

a composable function with tunable arguments $p \in \mathcal{Q}$, allowing flexible program construction tailored to different queries. The module set $\mathcal{M}$ spans a broad range of capabilities: retrieval-oriented modules such as GetClips and GetSubtitles return the top-$k$ relevant video segments or subtitle spans; temporal-control modules including TrimBefore, TrimAfter, and TrimRange enable precise interval selection; reasoning-support modules like QueryMC and QueryYN handle multiple-choice and yes/no questions, while RunOCR and DetectObject extract textual and object-level evidence from frames. Additional utilities such as GetSubsRange, GetCapsRange, and GetSubtitleHint provide finer-grained control over subtitle and caption data. By exposing reasoning as a sequence of function calls from $\mathcal{M}$, FS-VisPR treats video understanding as program execution rather than monolithic prediction. This design enhances interpretability and modularity. Compared with the previous design (Choudhury et al., 2023), we refine the vision module by achieving more precise subtitle localization, key frame extraction, and leveraging VideoLLM for improved reasoning. Full specifications of $\mathcal{M}$ and $\mathcal{Q}$ are provided in Appendix A.1.

### 3.3 DATASET CONSTRUCTION

We start from a training dataset $\mathcal{D}_{\text{train}} = \{(V_i, Q_i, A_i)\}_{i=1}^N$, where $V_i$ denotes a video, $Q_i$ a natural-language query, and $A_i$ the ground-truth answer. Based on this, we construct a visual program reasoning dataset $\mathcal{D} = \{(V_i, Q_i, A_i, P_i)\}_{i=1}^N$, where each $P_i$ is an executable visual program consisting of vision module plans and calls such that $\text{exec}(P_i) = A_i$. To generate $P_i$, we manually curate a small support set of samples $\mathcal{S} = \{(Q_s, A_s, P_s)\}$, which serve as few-shot prompts for a strong language model $p_\phi$. For each new instance, the model proposes a candidate program $\hat{P}_i \sim p_\phi(\mathcal{S}, Q_i)$, and we retain only those satisfying $\text{exec}(\hat{P}_i) = A_i$ as high-quality data. To annotate the query difficulty in the training set $\mathcal{D}_{\text{train}}$, we use the VideoLLM $p_\theta$ to obtain both the predicted answer and the associated confidence according to Eq. 1. $\hat{A}_i, \gamma_i = p_\theta(V_i, Q_i)$, and assign a difficulty label as

$$y_i = \begin{cases} \text{easy}, & \text{if } \hat{A}_i = A_i \text{ and } \gamma_i > \tau, \\ \text{difficult}, & \text{otherwise}, \end{cases} \quad \tau = 0.75. \tag{2}$$

The difficulty labels are embedded in the output prompts: "This is an easy question" triggers fast reasoning, while "This is a difficult question" invokes visual program reasoning. This supervision helps the model perceive query difficulty and adjust its reasoning strategy. To ensure balanced and diverse module usage, we track the frequency of each module in $\mathcal{D}$. If any module appears in fewer than 50 examples, we use $p_\phi$ for query rewriting to augment the dataset. As shown in Figure 3, the dataset enables FS-LLM to perform fast reasoning on easy queries and structured reasoning on difficult ones. Further details are in Appendix A.2.

### 3.4 PARAMETER SEARCH

To improve the robustness of program execution, FS-VisPR performs a parameter search over the module set $\mathcal{M}$ and associated parameter space $\mathcal{Q}$. Given a program $P = \{m_1(p_1), \dots, m_k(p_k)\}$, we vary one parameter $p_i \in \mathcal{Q}$ at a time to generate program variants $P'$, keeping all other arguments fixed from the original program. Multiple variants of the program can be run in parallel, enabling efficient exploration of different parameter configurations. During training, variants that yield the correct answer are retained to enrich the program space. During inference, the candidate program with the highest confidence result is applied. Key parameters include top-$k$ for retrieval modules, intervals for temporal modules, and num_frames for frame extraction. This procedure enables the model to systematically explore alternative module arguments, improving execution reliability without compromising modularity. Full details are provided in Appendix A.1.

### 3.5 ADAPTIVE FAST-SLOW REASONING

FS-VisPR integrates all components into an adaptive reasoning process. For a query $q$, the FS-LLM $p_\theta$ first predicts the difficulty label $y_i$ as either easy or difficult. Then, for easy queries, fast reasoning generates an answer and confidence $\gamma_i$. If $\gamma_i > c$, the answer is returned; otherwise, slow reasoning is invoked. For difficult queries, slow reasoning directly constructs a structured program from $\mathcal{M}$, executes it, and optionally explores parameter variations over $\mathcal{Q}$. The candidate with the highest confidence is selected. A fallback mechanism returns the fast reasoning prediction if program execution fails. This adaptive fast-slow process is shown in Figure 2 and Algorithm 1.

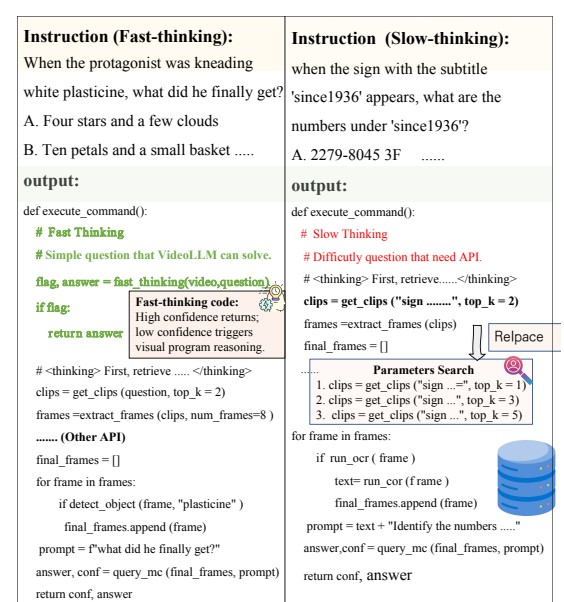

| Instruction (Fast-thinking): | Instruction (Slow-thinking): |
|---|---|
| When the protagonist was kneading white plasticine, what did he finally get? A. Four stars and a few clouds B. Ten petals and a small basket ..... | when the sign with the subtitle 'since1936' appears, what are the numbers under 'since1936'? A. 2279-8045 3F ...... |

Figure 3: Fast-Slow Dataset: enabling LLMs to adaptively choose between fast reasoning for simple queries, confidence-guided slow reasoning under uncertainty, and full slow reasoning for difficult queries as FS-LLM.

**Algorithm 1** Adaptive Fast-and-Slow Visual Program Reasoning

1: **Input:** FS-LLM $p_\theta$, video $v$, question $q$, vision modules $\mathcal{M}$, parameter set $\mathcal{Q}$, confidence threshold $\theta$
2: **Fast-slow reasoning:**
3:    $ot \leftarrow p_\theta(q, \texttt{stop} = \texttt{'return answer'})$
4: **if** $\texttt{'fast\_reasoning'}$ in $ot$ **then**
5:    $(confid, answer) \leftarrow \mathcal{M}[\texttt{fast\_think}](v, q)$
6:    **if** $confid \geq \theta$ **then**
7:      **return** $answer$    ▷ Return Fast-reasoning answer.
8:    **else**
9:      $ot \leftarrow p_\theta(q, ot)$    ▷ Trigger Slow-reasoning.
10:    **end if**
11: **end if**
12: **Execute the slow reasoning visual program:**
13:    $result \leftarrow \text{exec}(ot, \mathcal{M})$
14: **if fail:**
15:    **return** $\mathcal{M}[\texttt{fast\_think}](v, q)[\texttt{'answer'}]$
16: **Parameters Search:**
17:    $results \leftarrow [result]$
18: **for** each $p \in \mathcal{Q}$ **do**
19:    **for** each $value \in p$ **do**
20:      $output' \leftarrow \text{replace}(ot, p, value)$
21:      $results.\text{append}(\text{exec}(output', \mathcal{M}))$
22:    **end for**
23: **end for**
24: **return** $r.$answer
25: where $r = \arg\max_{r' \in results} r'.$confidence ▷ Return the answer corresponding to the result with the highest confidence

## 4 EXPERIMENTS

### 4.1 EXPERIMENTAL SETUP.

We construct the fast–slow reasoning dataset based on CG-Bench (Chen et al., 2024a), which contains 12K multiple-choice QA instances. Visual programs are generated using strong LLM, GPT-4.1 (OpenAI, 2025), and each query is labeled as *easy* or *difficult* according to confidence scores from Qwen2.5-VL (Bai et al., 2025a). The FS-LLM is fine-tuned on 5K diverse samples for three epochs with up to four NVIDIA H100 GPUs. Training and inference are conducted with LLaMAFactory (Zheng et al., 2024) and VLLM (Kwon et al., 2023), using Qwen3-8B (Yang et al., 2025; Hui et al., 2024) as the backbone for visual program code generation. For fast reasoning, Qwen2.5-VL supports the query_mc and query_yn modules, processing up to 64 frames. We evaluate FS-VisPR against both open-source baselines, LongVILALA (Chen et al., 2024b), VideoRAG (Ren et al., 2025), VideoMind (Liu et al., 2025), VideoXL (Shu et al., 2025), and Video-R1 (Feng et al., 2025), and proprietary models, GPT-4o (Hurst et al., 2024), Gemini 1.5 Pro (Team et al., 2024), and Seed 1.5VL-Pro (Guo et al., 2025). Benchmarks include LongVideoBench (Wu et al., 2024), VideoMME (Fu et al., 2024), and LVBench (Wang et al., 2024a). For LVBench, subtitles are generated from audio using FFmpeg and Whisper (Radford et al., 2023). All decoding follows the official configurations with the setting frames. The confidence threshold is by default 0.75. More dataset details are provided in Appendix A.3.

### 4.2 MAIN RESULTS

As shown in Table 3, FS-VisPR with a 7B VideoLLM achieves 50.4% on LVBench, surpassing GPT-4o (48.9%), 62.2% on VideoMME, slightly below Qwen2.5-VL-72B (64.6%) but above Qwen2.5-VL-7B (57.6%), and 62.2% on LongVideoBench, exceeding Qwen2.5-VL-72B (60.3%). Scaling to 34B as VideoLLM further improves results to 51.2%, 63.6%, and 64.8%. These results highlight FS-VisPR's consistent advantage over baseline VideoLLMs, its robustness across long-form video QA, and the benefit of modular, confidence-guided fast–slow reasoning. Table 4 analyzes the contribution

| Model | #Frame | LVBench | VideoMME (w sub, Long) | LongVideoBench | Avg |
|---|---|---|---|---|---|
| **Closed-source Models** | | | | | |
| GPT-4o (Hurst et al., 2024) | 384 | **48.9** | 72.1 | **66.7** | **62.6** |
| Gemini-1.5-pro (Team et al., 2024) | 256 | 33.1 | **77.4** | 64.0 | 58.2 |
| Seed1.5VL-pro (Team, 2025) | 32 | 46.1 | 63.3 | 63.7 | 57.7 |
| **Open-source Models** | | | | | |
| LongVILA-7B (Chen et al., 2024b) | 256 | - | 52.1 | 57.7 | - |
| LongVILA-7B + Video-RAG (Ren et al., 2025) | 32 | - | 55.7 | - | - |
| VideoMind-7B (Liu et al., 2025) | 2/FPS | 40.8 | 49.2 | - | - |
| Video-XL-7B (Shu et al., 2025) | 256 | - | 54.9 | 50.7 | - |
| Video-R1-7B (Feng et al., 2025) | 64 | - | 52.2 | - | - |
| InternVL3-8B (Zhu et al., 2025) | 64 | – | - | – | - |
| Qwen2.5VL-7B (Bai et al., 2025a) | 128 | 44.8 | 57.6 | 56.7 | 53.0 |
| Qwen2.5VL-7B-RAG (Bai et al., 2025a) | 128 | 47.2 | 58.2 | 58.4 | 54.6 |
| Qwen2.5VL-72B (Bai et al., 2025a) | 128 | 47.4 | **64.6** | 60.3 | 57.4 |
| FS-VisProgV-8B (w/ Qwen2.5VL-7B) | 64 | 50.4 | 62.2 | 62.2 | 58.3 |
| FS-VisProgV-8B (w/ Qwen2.5VL-34B) | 64 | **51.2** | 63.6 | **64.8** | **60.5** |

Table 3: Evaluation results on LVBench, VideoMME (long w/ subtitle), and LongVideoBench. FS-VisProg achieves the better performance among open-source methods and is competitive with closed-source models.

| Dataset | Reasoning | Samples | Short ↑ | Medium ↑ | Long ↑ | Avg ↑ | Output. Len ↓ | Avg. Runtime ↓ |
|---|---|---|---|---|---|---|---|---|
| VideoMME | Fast | 1611 | 85.1 | 79.8 | 71.2 | 79.3 | 366 | 2.4s |
| | Slow | 1089 | 43.6 | 49.1 | 52.0 | 48.9 | 1350 | 6.2s |
| | Overall | 2700 | 73.1 | 66.2 | 62.1 | 62.2 | 762 | 3.9s |
| LongVideoBench | Fast | 737 | 83.7 | 80.7 | 76.5 | 80.4 | 366 | 2.8s |
| | Slow | 598 | 51.3 | 50.0 | 44.3 | 47.4 | 1696 | 6.8s |
| | Overall | 1335 | 70.5 | 64.0 | 55.5 | 62.2 | 964 | 4.6s |

Table 4: Accuracy, output length, average runtime, and sample distribution of **fast**, **slow** reasoning across video durations. Fast reasoning corresponds to direct responses from the VideoLLM, while slow reasoning arises either when the model directly generates a visual program for perception or when fast reasoning yields low confidence, triggering a second-stage inference. Most samples are resolved by fast reasoning (e.g., 1611/2700 on VideoMME and 737/1335 on LongVideoBench).

of fast and slow reasoning. Fast reasoning corresponds to direct VideoLLM answers, achieving strong accuracy on short videos (e.g., 85.1% on VideoMME) with concise outputs (366 length on average), but degrading on longer videos. Slow reasoning is triggered either when the model directly generates a visual program or when fast reasoning falls below the confidence threshold, producing longer programs (up to 1696 length) while maintaining stable accuracy around 50%, and computed the average runtime per sample, including the time for generating and executing the visual program, to demonstrate the efficiency of FS-VisPR. We also present the results obtained using VideoLLM (as Fast-thinking) for the 598 slow-reasoning samples in LongVideoBench and the 1,089 samples in VideoMME in Appendix A.4.

## 4.3 ABLATION STUDIES

**Confidence Threshold**  We examine the effect of the confidence threshold $\theta$, which determines whether the model accepts an answer from fast reasoning or switches to slow reasoning. As shown in Figure 4, increasing $\theta$ from 0.4 to 0.75 yields an accuracy gain of over 3%. However, further raising the threshold leads to performance degradation. A low threshold ($\theta = 0.4$) causes the model to rely predominantly on fast reasoning, whereas a high threshold ($\theta = 0.9$) shifts most decisions to slow reasoning. These results suggest that fast reasoning is effective for simple queries, while slow reasoning is better suited for difficult ones. Overall, adaptively choosing between fast and slow reasoning based on confidence surpasses either strategy alone.

**Visual Modules**  Table 6 reports the impact of different module settings on FS-VisPR performance for LongVideoBench and LVBench. For retrieval modules, increasing Top-$k$ improves accuracy, with LongVideoBench peaking at 61.2 for Top-$k = 3$ and LVBench at 49.2 for Top-$k = 5$, while omitting retrieval reduces accuracy to 57.8 and 46.3, respectively. Temporal control via `Trim` achieves the best accuracy on LongVideoBench at 60.7 with 30-second intervals, whereas

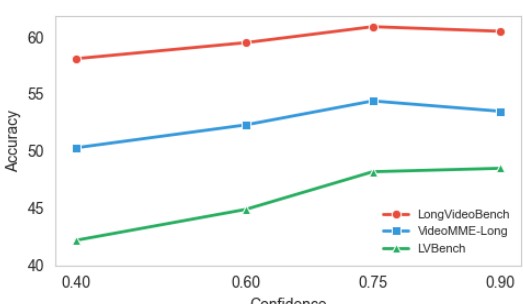

| Parameter Set | Voting | Conf. (Ours) |
|---|---|---|
| None | 44.7 | 44.7 |
| Num_frames | 44.5 | 44.9 |
| Intervals | 44.4 | 44.4 |
| Top_k | 44.5 | 46.2 |
| Num_frames + Top_k | 43.8 | 46.9 |
| Num_frames + Intervals | 44.2 | 46.8 |
| All | 44.4 | **47.1** |

Table 5: The Parameter Search results on LongVideoBench slow-reasoning data. Voting selects the most frequent prediction; Confidence selects the prediction with the highest confidence. "None" indicates no parameter search integration.

Figure 4: Performance of the FS-VisPR at varying confidence threshold on Long-form VideoQA.

| Dataset | Modules | Setting Parameter | | | | | |
|---|---|---|---|---|---|---|---|
| LongVideoBench | Retrieval | Top-$k$ | *N/A* | 1 | 3 | 5 | — |
| | | Acc. (%) | 57.8 | 60.2 | **61.2** | 60.1 | — |
| | Trim | Intervals | *N/A* | 10 | 20 | 30 | 60 |
| | | Acc. (%) | 58.7 | 60.4 | 60.4 | **60.7** | 60.5 |
| | Detect_object | Text-thr | *N/A* | 0.25 | 0.50 | 0.70 | — |
| | | Acc. (%) | 59.8 | **60.9** | 60.4 | 60.4 | — |
| | Extract_frames | Num_frames | 8 | 16 | 32 | 64 | — |
| | | Acc. (%) | 58.2 | 59.5 | 60.2 | **60.8** | — |
| LVB | Retrieval | Top-$k$ | *N/A* | 1 | 3 | 5 | — |
| | | Acc. (%) | 46.3 | 46.7 | 48.8 | **49.2** | — |
| | Extract_frames | Num_frames | 8 | 16 | 32 | 64 | — |
| | | Acc. (%) | 44.8 | **46.4** | 44.2 | 45.8 | — |

Table 6: Ablation study of different **Modules** on **LongVideoBench** and **LVB**. For each Module and parameter, the first row lists parameter values, while the second row reports the corresponding accuracy. The "N/A" label indicates that the Module is not activated in visual program reasoning.

disabling trimming lowers performance to 58.7. For object detection, adjusting the text threshold (`Text-thr`) provides modest gains (60.9), while skipping detection reduces accuracy to 59.8. Increasing the number of extracted frames consistently benefits LongVideoBench (up to 60.8), though improvements on LVBench are mixed; removing frame extraction lowers LongVideoBench accuracy to 58.2. These findings highlight the contribution of each Module and underscore the importance of proper parameter settings for robust long-form VideoQA.

**Module Parameter Search** We evaluate the module parameter search on **LongVideoBench** (1,089 slow-reasoning samples). As shown in Table 5, varying Num_frames (8, 16, 32, 64) from the ExtractFrames, and `Top_k` from the GetClips (1, 3, 5) yields seven program variants under the "Num_frames + Top_k" setting, while the "All" configuration integrates all tested parameter sets. For aggregation, majority voting selects the most frequent prediction, whereas confidence-based selection chooses the prediction with the highest confidence. Confidence consistently outperforms voting (e.g., 46.9 vs. 43.8 on "Num_frames + Top_k"), and for single-parameter searches, Top_k proves more effective compared to Num_frames and Intervals, achieving a result of 46.2. The "All" configuration achieves the best accuracy of 47.1, representing a 2.4 improvement over the 44.7 result without parameter search integration. These results demonstrate that combining diverse parameter configurations enhances both the robustness and accuracy of visual program reasoning.

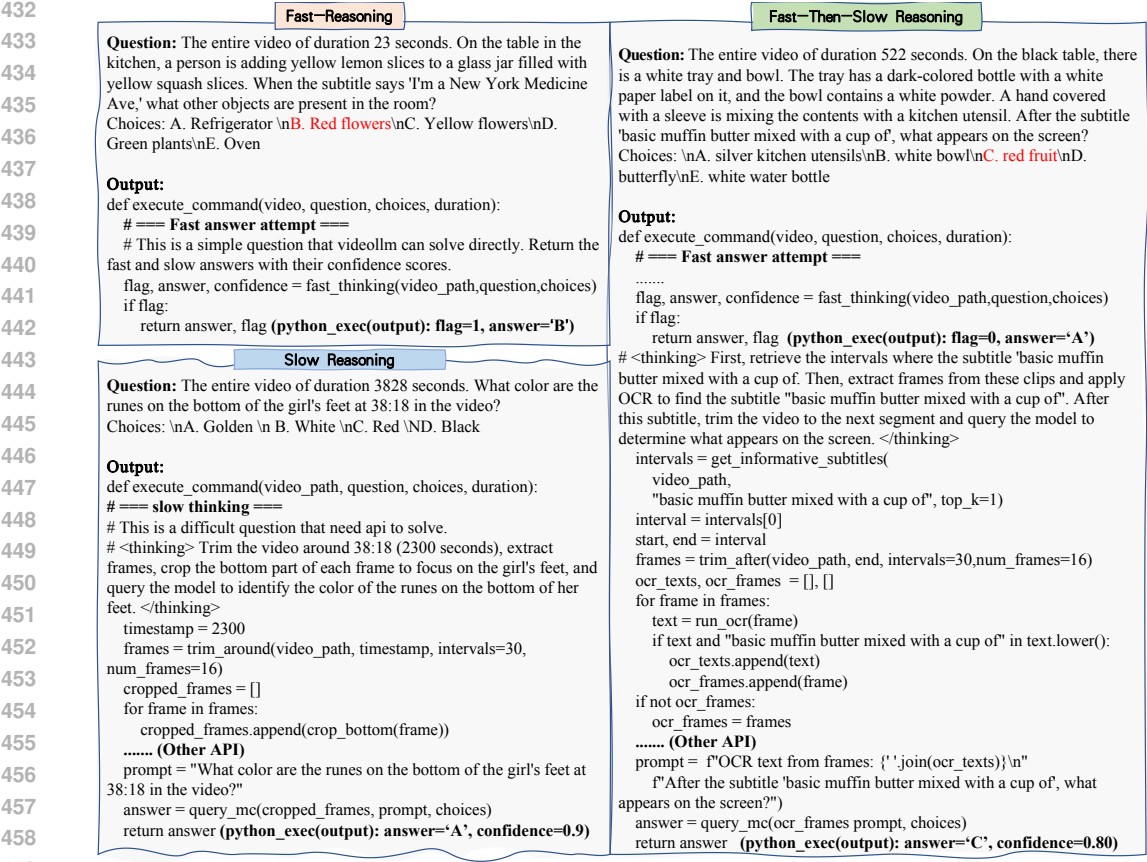

Figure 5: FS-VisPR case study: Three primary reasoning modes: direct fast reasoning via Vide-oLLM, slow reasoning through generated visual programs, and slow reasoning triggered when fast reasoning yields low confidence.

## 4.4 CASE ANALYSIS

We present representative cases from Figure 8 to illustrate FS-VisPR's adaptive, query-aware reasoning. For a short 23-second video, the model uses fast reasoning to correctly answer a simple object query. In a long 3828-second video, slow reasoning is required: FS-VisPR perceives the query's difficulty, locates the timestamp, extracts and crops frames, and identifies the correct answer, highlighting both temporal-spatial precision and interpretability via visual programs. In two-stage inference, low-confidence fast reasoning triggers slow reasoning with frame retrieval and OCR to obtain the correct result. All visual programs employ parameter search to select the highest-confidence outcome, providing transparent and explainable reasoning steps. These examples demonstrate that FS-VisPR not only adapts its strategy based on perceived query difficulty but also delivers interpretable, program-based evidence. We also show some failure cases in Appendix A.5.

## 5 CONCLUSION

In this work, we introduced FS-VisPR, an adaptive visual program reasoning framework for long-form VideoQA. Leveraging model confidence as a reliable signal, FS-VisPR dynamically balances fast reasoning for simple queries with slow, program-based reasoning for difficult queries. By generating explicit visual programs, the framework enhances interpretability, allowing users to understand how answers are derived. We developed a set of efficient modules, constructed a fast-slow aligned dataset, and proposed a parameter search mechanism to improve program diversity and robustness. Extensive experiments demonstrate that FS-VisPR achieves effective, adaptive, and interpretable video question answering, outperforming existing VideoLLMs.

ETHICS STATEMENT

This work adheres to the ICLR Code of Ethics. No human subjects or animal experimentation were involved in this study. All datasets used, including CG-Bench (Chen et al., 2024a), VideoMME (Fu et al., 2024), LongVideoBench (Wu et al., 2024), and LVBench (Wang et al., 2024a), were sourced in accordance with the relevant usage guidelines, ensuring compliance with privacy standards. We have made efforts to prevent any biases or discriminatory outcomes throughout our research. No personally identifiable information was used, and no experiments were conducted that could raise privacy or security concerns. We remain committed to upholding transparency and integrity throughout the research process.

REPRODUCIBILITY STATEMENT

We have made every effort to ensure that the results presented in this paper are reproducible. We introduce a pipeline for dataset generation in Section 3.3, which outlines the construction of the slow-fast reasoning data. A comprehensive description of the pipeline, including the specific prompts used, is provided in Appendix A.2. For evaluation, baseline models and their configurations, as well as the training strategies and datasets employed to enhance model performance, are discussed in Section 4.1. We believe these measures will enable other researchers to reproduce our work and further advance the field.

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

# A APPENDIX

## A.1 MODULE DETAILS

We design a suite of modules to support video understanding and reasoning. Each module is formally specified by its parameters and parameter search space as follows.

- **GetClips**
  - **Parameters**: `video_path`, `query`, `top_k`
  - **Description**: Segments the video into 10-second clips, encodes each with `LanguageBind_Video` (Zhu et al., 2023), and retrieves the top-$k$ clips most relevant to the query.
  - **Parameter Search**: `top_k` $\in \{1, 3, 5\}$

- **GetSubtitles**
  - **Parameters**: `video_path`, `query`, `top_k`
  - **Description**: Retrieves subtitles (generated by Whisper since LVBench lacks transcripts) (Zhu et al., 2023) and returns the top-$k$ segments aligned with the query and their timestamps.
  - **Parameter Search**: `top_k` $\in \{1, 3, 5\}$

- **TrimBefore**
  - **Parameters**: `video_path`, `timestamp`, `intervals`
  - **Description**: Removes all content before the given `timestamp`, retaining only the following `intervals`.
  - **Parameter Search**: `intervals` $\in \{10, 20, 30, 60\}$

- **TrimAfter**
  - **Parameters**: `video_path`, `timestamp`, `intervals`
  - **Description**: Removes all content after the given `timestamp`, retaining only the preceding `intervals`.
  - **Parameter Search**: `intervals` $\in \{10, 20, 30, 60\}$

- **TrimRange**
  - **Parameters**: `video_path`, `start`, `end`
  - **Description**: Extracts the segment between the `start` and `end` timestamps.
  - **Parameter Search**: `intervals` $\in \{10, 20, 30, 60\}$

- **QueryMC**
  - **Parameters**: `frames`, `query`, `choices`
  - **Description**: Answers multiple-choice questions using videoLLM and Qwen2.5-vl (7B/34B), returning both the predicted answer and confidence score.
  - **Parameter Search**: None

- **QueryYN**
  - **Parameters**: `frames`, `query`
  - **Description**: Answers binary yes/no questions using videoLLM and Qwen2.5-vl (7B/34B).
  - **Parameter Search**: None

- **RunOCR**
  - **Parameters**: `frame`
  - **Description**: Performs OCR on the input frame using EasyOCR and returns the recognized text.
  - **Parameter Search**: None

- **DetectObject**
  - **Parameters**: `frame`, `text`, `text_thr`, `box_thr`

- **Description**: Detects objects in a frame conditioned on a textual query using DEVA (Cheng et al., 2023), and outputs bounding boxes.
  - **Parameter Search**: `text_thr` and `box_thr` control detection thresholds.

- **GetSubsRange**

  - **Parameters**: `video_path`, `start`, `end`
  - **Description**: Retrieves subtitle segments between `start` and `end`.
  - **Parameter Search**: None

- **GetCapsRange**

  - **Parameters**: `video_path`, `start`, `end`
  - **Description**: Retrieves captions within the specified range, similar to **GetSubsRange**.
  - **Parameter Search**: None

- **GetSubtitleHint**

  - **Parameters**: `video_path`, `query`
  - **Description**: Provides query-based subtitle hints using Qwen3-8B (Yang et al., 2025).
  - **Parameter Search**: None

- **Crop**

  - **Parameters**: `frame`, `box`
  - **Description**: Crops the specified region of interest from a frame for fine-grained analysis.
  - **Parameter Search**: None

- **ExtractFrames**

  - **Parameters**: `video_path`, `num_frames`
  - **Description**: Extracts frames from the video for subsequent processing.
  - **Parameter Search**: `num_frames` controls the number of sampled frames.

- **SplitVideo**

  - **Parameters**: `video_path`
  - **Description**: Splits the video into candidate intervals based on scene boundaries for fine-grained analysis using `scenedetect` package.
  - **Parameter Search**: None

## A.2 PROMPTS

We provide GPT-4.1 few-shot prompts to generate visual reasoning programs for CG-Bench, including the reasoning (*thinking*) process and module workflows, as shown in Figure 6. We also supply code for question and workflow refinement to diversify module usage in the training data, illustrated in Figure 7.

## A.3 DATASETS

LongVideoBench (Wu et al., 2024) is a benchmark for long video understanding, containing 3,763 videos (up to 1 hour) with subtitles and 6,678 human-annotated multiple-choice QA pairs across 17 categories. VideoMME (Fu et al., 2024) provides 900 videos (254 hours) with 2,700 human-curated QA pairs across six domains and 30 subcategories. Videos range from 11 seconds to 1 hour and include frames, subtitles, and audio to evaluate multimodal reasoning. LVBench (Wang et al., 2024a) targets extreme long video comprehension, featuring videos from 70 seconds to 4 hours. It covers single-scene, multi-scene, and full-scene settings with diverse reasoning types, such as temporal, spatial, causal, hypothetical, and external knowledge.

```
<Prompt>
You are a video reasoning agent. Your task is to analyze a video question and generate a program with available
APIs.  ## API Reference:
retrieval functions:
   get_informative_clips(query, video_path, top_k=3, total_duration=duration)  # Returns intervals, files (time
intervals and file paths)
   get_informative_subtitles(query, video_path, top_k=1, total_duration=duration)  # Returns subtitles list
analysis_manager functions:
   query_mc(frames, query, choices)  # Answers a multiple-choice question using the frames.
   query_yn(frames, query)  # Answers yes/no questions.
   trim_after(video_path, timestamp)  # Trims the video after the timestamp.
   trim_before(video_path, timestamp)  # Trims the video before the timestamp.
   trim_frames(video_path, start_timestamp, end_timestamp)  # Trims the video between given timestamps.
   run_ocr(frame)  # Performs OCR on a frame and returns recognized text.
   detect_object(frame, text)  # Detects an object in a frame based on text, returns bounding boxes.
   crop(frame, box)  # Crops the given region in a frame.
other functions:
   extract_frames(video_path)  # Extracts frames from the video.
   split_video(video_path) # Split the video into candidate temporal intervals.
   get_subtitles_in_range(video_path, start_timestamp, end_timestamp)  # Gets subtitles in the specified range.
   get_captions_in_range(video_path, start_timestamp, end_timestamp)  # Gets captions in the specified range.
Allowed Utility Functions:
   Use standard Python constructs like if, for, len(), max(), sorted(), etc.
---
Analyze the video question and generate Python code using the APIs.
Your output must define a Python function in the following format:

def execute_command(video_path, question, choices, duration):
   # Step-by-step reasoning using available APIs, with the <thinking> </thinking> tag.
   # The visual program code.
   ...
   return result
]
Here are some examples:
<Few-shot examples>
```

Figure 6: The GPT-4.1 prompt used to generate visual program reasoning data.

### A.4   MORE RESULTS

As shown in Table 7, the results based on slow reasoning (1,089 samples from VideoMME and 598 samples from LongVideoBench) are compared with the results obtained using fast reasoning on the same samples. In both datasets, the performance of fast reasoning is consistently lower than that of

<Prompt>

Keep the underlying logic, scenario, and expected action consistent with the example, but you can freely vary any aspect such as wording, descriptions of characters or objects, their positions, appearance, clothing, location, background elements, or any other relevant detail. You may also rephrase options, change their order, or introduce reasonable variations, as long as the main reasoning and correct answer type remain valid. The goal is to produce diverse, natural, and logically consistent new examples.

Example:
Instruction:
{example['instruction']}

Output:
{example['output']}

Now generate one new example following the same format (Instruction + Output).
{generalized_instruction}

Figure 7: The GPT-4.1 prompt used to rewrite queries and workflows to generate diverse module data.

| Dataset | Type | Short ↑ | Medium ↑ | Long ↑ | Avg ↑ |
|---------|------|---------|----------|--------|-------|
| VideoMME | Fast | 38.3 | 42.2 | 36.8 | 39 |
| | Slow | 43.6 | 49.1 | 52.0 | 48.9 |
| LongVideoBench | Fast | 47.2 | 39.3 | 36.4 | 38.3 |
| | Slow | 51.3 | 50.0 | 44.3 | 47.4 |

Table 7: Most results are based on the 1,089 slow-reasoning samples from VideoMME and the 598 slow-reasoning samples from LongVideoBench, compared with the fast reasoning results.

slow reasoning across all categories (*Short*, *Medium*, *Long*, and *Avg.*). These comparisons demonstrate that slow reasoning yields better results, especially when handling more complex queries where fast reasoning tends to perform less effectively.

## A.5 CASE STUDIES: FAILURE ANALYSIS

In addition to successful cases, we highlight two classes of failure cases. First, when the VideoLLM is overconfident in an incorrect answer, confidence-based triggering may fail to activate slow reasoning. This limitation could be mitigated by adopting stronger baseline VideoLLMs with improved calibration. Second, there exist queries that cannot be resolved even with slow reasoning. Such cases point to the need for expanding the advanced module library, underscoring the extensibility of the FS-VisPR framework.

## A.6 USAGE OF LLMS IN PAPER WRITING

Large Language Models (LLMs) were used to aid in the writing and polishing of the manuscript. Specifically, we used an LLM to assist in refining the language, improving readability, and ensuring clarity in various sections of the paper. The model helped with tasks such as sentence rephrasing, grammar checking, and enhancing the overall flow of the text.

It is important to note that the LLM was not involved in the ideation, research methodology, or experimental design. All research concepts, ideas, and analyses were developed and conducted by the authors. The contributions of the LLM were solely focused on improving the linguistic quality of the paper, with no involvement in the scientific content or data analysis.

The authors take full responsibility for the content of the manuscript, including any text generated or polished by the LLM. We have ensured that the LLM-generated text adheres to ethical guidelines and does not contribute to plagiarism or scientific misconduct.

**Fast Reasoning**

The entire video of duration 3665 seconds. Question: What does the protagonist see through the window after she is taken to the utility room?
(A) A group of monks sitting cross-legged in the snow
(B) A group of citizens chatting together
(C) A group of warriors practicing swords
(D) A group of samurais eatingOutput:
def execute_command(video, question, choices, duration):
    # === **Fast answer attempt** ===

**Overconfidence in VideoLLMs**

    .......
    flag, answer, confidence = fast_thinking(video_path,question,choices)
    if flag:
        return answer, flag  **(python_exec(output): flag=1, answer='A')**

**Slow Reasoning**

**Question:** The entire video of duration 2548 seconds.
Question: How many doors have colored lights at 00:29?Choices: \nA. 2 \n B.1 \nC.3\ND.0
def execute_command(video_path, question, choices, duration):
# === slow thinking ===
<think>
......
</think>


def execute_command(video_path, question, choices, duration):
# === slow thinking ===
# This is a difficult question that need api to solve.
# <thinking> Trim the video around 29 seconds to focus on the relevant scene.
    # Extract frames and detect doors with colored lights in each frame.
    # Count the number of such doors per frame, and use the maximum count from all frames.
    # Finally, query the model to select the correct answer from the choices.
</thinking>
    timestamp = 29
    frames = trim_around(video_path, timestamp, intervals=16, num_frames=16)
    **....... (Other API)**
    prompt = (
        f"Question: How many doors have colored lights at 00:29?"
    )
    answer = query_mc(frames, prompt, choices)
    return answer
  **(python_exec(output): confidence=0.94, answer='B')**

Figure 8: Error case study: VideoLLM errors caused by overconfidence and limited counting ability in the query_mc Module.

