# OpenReview forum: "Adaptive Fast-and-Slow Visual Program Reasoning for Long-Form VideoQA"
_ICLR.cc/2026/Conference — ICLR 2026 Conference Withdrawn Submission_

### Official Review · Reviewer_rdU6 · 2025-10-17

**Soundness:** 3
**Presentation:** 3
**Contribution:** 2
**Rating:** 2
**Confidence:** 4

**Summary:**

This paper proposes a neural-symbolic approach for language-based slow-fast reasoning. This paper designs several efficient visual modules, such as key clip retrieval and subtitle retrieval for long-form videos. This paper then constructs a diverse and high-quality fast-slow reasoning dataset. Experiments show that they can achieve stronger performance than GPT-4o on LVBench.

**Strengths:**

1. The experimental results look promising as it outperforms several competitive baselines such as QWEN and longvila.

2. We can observe that the proposed 8B model even outperforms the 72B model of Qwen, which is quite amazing.

**Weaknesses:**

1. The central claim of the evaluation results is that it outperforms open-source models. But in fact, this work isn't open-sourced. The attached supplementary code seems to be irrelevant to this submission and is on a toy dataset. The code is also specific in different situations. These facts raise overall concerns about the quality of this work.

2. From a methodology perspective, there are several things important to their contribution, but not clear:

(1) In Table 2, what are introduced by this work, and what are the existing methods? I think from VisualProg and follow-up works, these operators are mostly ready to use.

(2) How can the modules be connected if they have very diverse signatures? Is there any syntax checking algorithm?

(3) How can each module's effectiveness be justified? Note that Table 6 only ablates some parameters of some modules, while not all modules are justified. Also, the result difference of module ablations in Table 6 is quite small.

 (4) Is the parameter set in each module related to specific datasets? Those parameters may not be generalizable.

The main concern is that the claimed contribution #1 is weak and it's not novel, given so many related neural symbolic works.

(5) The training parameters are not described clearly. Ln 257 says "during training", it's not clear what it refers. Parameter search is not typically called a training procedure. This part needs to be clarified.

3. About the dataset construction. Is the dataset of 12K only enough to train the model? It's unclear whether the benefit comes from the training dataset or the model design itself. Also, it's not certain whether the data might contain samples close to the testing dataset.

**Questions:**

1. Why does InternVL have no results numbers listed? If so, why list it in Table 3?

2. Why not set the full column-width of Table 3?

3. Methodology questions: see above weakness section.

---

> ### Author Response · Authors · 2025-11-13
>
> ## Weakness 1: Submission of Supplementary Materials
>
> We apologize for the oversight in submitting the supplementary materials. The correct source code will be updated and included in the supplementary materials. Furthermore, we will make the code publicly available on **GitHub** to promote research in the field of long video understanding. We hope that this will facilitate further advancements and collaboration within the community.
>
> ---
>
> ## Weakness 2: Methodology Clarifications
>
> ### 1. Long Video Processing Modules
>
> Previous work, such as **VisualProg**, primarily focused on visual tasks and short video QA. The modules in these works are not optimized for handling long videos. In contrast, our approach to long video processing involves the following:
> - **Splitting the video** into 10-second clips
> - Using **Whisper** for **audio-subtitle extraction**
> - Leveraging **LanguageBind** to create **multimodal embeddings** for video frames and subtitles
> - Integrating **VideoLLM** into a query API to provide **confidence scores** and **predicted answers**, thus enhancing the reasoning process.
>
> These techniques for handling long videos are novel and have not been explored in previous works.
>
> ---
>
> ### 2. Connecting Modules with Diverse Signatures
>
> Our framework connects modules with diverse signatures through a **Python program**. One module's output can serve as the input to another. We design a **few-shot approach** with the help of **GPT-4.1** to generate high-quality and diverse visual programs. This process involves **parameter search** for each module, allowing us to create variants of visual programs by adjusting the parameters. For example, the **"replace operator"** in **Figure 3** demonstrates how this process works.
>
> ---
>
> ### 3. Justifying the Effectiveness of Each Module
>
> Each tool module in our system is intuitively designed for specific query scenarios. For instance:
> - **Trim**: Used when asked, "What happened at the beginning of the video?" Trimming the frames at the start of the video is a reasonable approach.
> - **GetClips**: Useful for questions like, "What is the person doing when the red table appears in the video?" By locating relevant clips, we provide context for the query.
> - **GetSubtitles**: Works with audio-subtitles for text-based queries.
>
> These modules are designed based on video scene understanding, and we validated their effectiveness through extensive **ablation studies**. **GetClips** showed performance fluctuations, as detailed in **Table 6**.
>
> ---
>
> ### 4. Parameter Sets and Dataset Generalization
>
> The parameters within each module (e.g., **top_k** for **GetClips**) are **not specific to the dataset**, but are tied to the design of the vision modules. For example:
> - **top_k** controls how many clips are retrieved for a given query.
> - These parameters are intended to enhance the module's flexibility and performance, and they are independent of the specific dataset used.
>
> We hope this clarifies the nature of the parameters and their role in module design.
>
> ---
>
> ### 5. Clarification of Training Procedure
>
> As explained in **Figure 2**, the training phase includes:
> - **Data construction** and **model fine-tuning**
> - During training, visual programs are generated by varying parameters (e.g., **top_k** for **GetClips**) with values like **top_k=3**, **top_k=5**, and **top_k=1**. These variations generate different programs based on the number of clips retrieved.
> - The best-performing programs (those yielding correct answers) are retained as training data to fine-tune the LLM.
>
> This process allows us to refine the model and improve its performance.
>
> ---
>
> ## Questions
>
> ### 1. Why does InternVL have no results listed? If so, why list it in Table 3? Why not set the full column-width of Table 3?
>
> Thank you for pointing this out. The omission of results for **InternVL** in **Table 3** was an oversight. We will correct this in the next version and ensure it is properly addressed and clarified.
>
> Additionally, we will adjust the table formatting in the next version to ensure that the column width is set correctly, making the data fully visible and easier to interpret.
>
> ---
>
> We sincerely appreciate the reviewer’s constructive feedback and will incorporate these changes into the next version. We hope that these clarifications will address your concerns and lead to a more favorable evaluation of our work.

---

> > ### Comment · Reviewer_rdU6 · 2025-11-13
> >
> > Thank you for your rebuttal and for your planned revisions to address those oversights. I am looking forward to reviewing them.
> >
> > My final recommendation will be made after the reviewer discussion period.

---

> > > ### Author Response · Authors · 2025-11-13
> > >
> > > Thank you very much for your careful reading and constructive feedback. We sincerely appreciate your willingness to revisit the submission after the discussion period.
> > >
> > > To briefly reiterate our contributions: we design long-video visual modules and construct two types of visual-program reasoning using GPT-4.1 as a code-generation LLM to fine-tune open-source models into FS-LLM; we introduce adaptive fast–slow reasoning at inference time (Algorithm 1), propose robustness strategies including confidence-based routing and fallback reasoning, and employ parameter search to further improve long-video program reliability. Our framework ultimately achieves performance comparable to state-of-the-art large-scale VideoLLMs in the long-form video benchmark.
> > >
> > > Thank you again for your time and thoughtful review.

---

### Official Review · Reviewer_2xFo · 2025-10-21

**Soundness:** 3
**Presentation:** 3
**Contribution:** 2
**Rating:** 4
**Confidence:** 4

**Summary:**

This paper proposes FS-VisPR a Fast-and-Slow Visual Program Reasoning for Long-Form VideoQA framework, the proposed method  consists of an adaptive visual program reasoning approach, that depending on the difficulty of the input question, the pipeline is able to adapt and choose "Fast Reasoning" for simple input queries, where a single VLM is used to give a direct answer, on the other hand, if the query is more complex, the pipeline uses "Slow Reasoning" by using visual program reasoning - a sequence of vision modules to gather evidence and answer the question -. To train the proposed model the authors construct a new dataset, that takes existing VQA datasets as a starting point and augment them with visual programs. FS-VisPR achieves better performances than Open-Source Models on LVBench, VideoMME and LongVideoBench and is competitive with Close-source models such as GPT-4o, Gemini-1.5-pro and Seed1.5VL-pro.

**Strengths:**

Paper Strengths:
1)  FS-VisPR a method that couples fast and slow reasoning for long VideoQA. The proposed approach demonstrates a practical recipe to tackle long-videoQA with test-time compute adaptively, that also provides interpretability from visual programs and confidence outputs.
2) A data construction pipeline that labels questions depending on their difficulty, and helps for aligning an openLLM to emit either a direct answer "fast reasoning" or start a visual program for "slow reasoning".
3) Broad benchmarking on LVBench, VideoMME, LongVideoBench with competitive or superior accuracy compared with open and close source models.
4) The paper is well written and easy to follow.

**Weaknesses:**

Paper Weaknesses:
1) The main comparison done in Table 3, is evaluating the proposed model FS-VisPR which uses fast and slow reasoning against single-pass videoLLMs for example against the backbone model Qwen2.5VL-7B, so improvements partly reflect extra test-time compute, not just a stronger backbone. Slow reasoning (which I would say is similar to chain-of-thought) is helping to get better performances, therefore, I think the comparison is not really fair. Have you tried to prompt the other models to perform chain of thought?
2) FS-VisPR uses FS-LLM (Qwen3-8B) to decide easy vs difficult, which is trained without any direct visual signal for a task that is related to video. This can create a suboptimal routing which can waste time. On the other hand,  FS-LLM does no see raw video while planning, from my understanding it writes programs from the question only, which I think can lead to noisy learning or suboptimal plans given by the LLM. Could this also mean that the LLM could be leveraging the well know problem in VQA datasets, which a model can reach the answer without seeing the visual context?.
3) I think is difficult to check the plans created by the model with the current Figures. The paper could benefit with some Figure showing this in a more visual way, for example if the model selected frames or OCR was used, it would be good to have some examples where you can see this visually , it could help to get a better understanding in my opinion.
4) While the paper motivates long videos and evaluates on long-form benchmarks, its analysis of length effects is limited to coarse Short/Medium/Long bins in a table. For example, in my opinion a curve of average accuracy vs video length can help to clarify the performance of the model and created a more focused analysis in long videoQA.
5) I think there is a typo on section 4.4 , maybe it should be Figure 5 and not Figure 8.

**Questions:**

Please refer to weaknesses for my questions and doubts. Overall, I think the paper brings a nice idea to use a fast-slow reasoning framework for long VideoQA, however I want to see the clarifications to my questions and concerns and sorry if I misunderstood something, I look forward to see the author's responses. Currently, I'm giving borderline reject, after rebuttal I will revise my decision.

---

> ### Author Response · Authors · 2025-11-13
>
> ## Weakness 1: Fair Comparison of Fast and Slow Reasoning
>
> We understand the reviewer’s concern regarding the fairness of comparing **FS-VisPR** with a single-pass **VideoLLM** (such as **Qwen2.5VL-7B**). To clarify, **FS-VisPR** employs two reasoning strategies: **fast reasoning** and **slow reasoning**.
>
> - **Fast reasoning** uses **VideoLLM** with a uniform frame sampling strategy to quickly generate predictions.
> - **Slow reasoning** leverages a **visual program** (such as key-frame retrieval modules) to locate key information, which is then fed back into **VideoLLM** for further inference, as shown in **Figure 5**.  ((the last vision module in visual program is Query_mc, which will  input the key frames and effective text and call the VideoLLM to get the answer and confidence.)
>
> For example, in the case of a query like "After action A in the video, what happens to person B?", we first use **GetClips** to locate the time interval for action A. Then, using **Trim_after**, we extract frames from the interval following action A, and use **OCR** to filter the person in the frame. Finally, these frames are input into **VideoLLM** for reasoning. This visual program approach helps pinpoint relevant parts of the video before feeding them into the model, ensuring that **slow reasoning** effectively adds value.
>
> When comparing to **Video-r1**, which outputs predictions along with "//<thinking//>" and //<answer//>, **FS-VisPR** shows a performance improvement from **52.2% to 62.2%** on the **VideoMME-L** testset  in table 3, indicating the effectiveness of the fast-slow reasoning framework.
>
> ---
>
> ## Weakness 2: Suboptimal Routing with FS-LLM (Qwen3-8B)
>
> We acknowledge the reviewer’s concern about **FS-LLM (Qwen3-8B)** being trained without direct visual signals for video-related tasks, which could potentially lead to suboptimal routing in some cases. The model is indeed designed to focus on the **query’s characteristics** and decide whether a visual program is needed.
>
> To address this, we introduce an additional input, **video_duration**, to **FS-LLM**, which helps guide the model to lean towards **slow reasoning** when faced with long videos. This ensures that **FS-LLM** can adapt better when the context of a long video is involved. Previous works like **VisProg**, **ViperGPT**, and **ProViQ** also utilized closed-source GPT models for generating visual programs using few-shot learning, highlighting that **LLMs** can effectively generate programs based on query inputs.
>
> Moreover, we have designed robust strategies and a **parameter search mechanism** to further improve the system’s performance. Specifically:
> 1. (1)**VideoLLM confidence** is used to decide whether to invoke visual program reasoning or (2) fall back to intrinsic **VideoLLM** reasoning when the program generation fails.
> 2. We introduce a **parameter search** during training and inference, inspired by human hyperparameter tuning. By varying parameters like **Top_k** = {1, 3, 5}, we generate multiple candidate programs. During training, only programs that yield correct answers are retained, and during inference, the program with the highest confidence score is applied.
>
> We also want to clarify the reviewer’s concern about leveraging the **VQA dataset** issue, where models can sometimes reach answers without the visual context. In contrast, previous works like **VisProg**, **ViperGPT**, and **ProViQ** also utilized GPT models for generating visual programs using few-shot learning. Importantly, **visual programs** are executed in a way that requires **visual context** to solve the task. Similarly, in our method, the visual program ensures that the model sees the necessary visual context before drawing conclusions.
>
> ---
>
> ## Weakness 3, 4, and 5: Analysis of Length Effects and Presentation
>
> We appreciate the reviewer’s suggestion to enhance the analysis of **video length effects**. Currently, we categorize videos into **short (<2 min)**, **medium (2–15 min)**, and **long (>30 min)**. However, we understand that a more detailed analysis could be beneficial, such as a curve showing **accuracy vs. video length**. In the next version, we will update the presentation to provide a more granular breakdown of how **FS-VisPR** performs with videos of varying lengths.
>
> Additionally, we will correct the **error in Section 4.4**, where it incorrectly refers to **Figure 8** instead of **Figure 5**. We apologize for this oversight and will update it in the next version.
>
>
> Thank you for your constructive feedback. We will update the presentation and clarify the points raised to ensure the methodology and results are clearer and more robust. We hope the revisions will address your concerns and improve the overall evaluation of our work. We appreciate your thoughtful review and look forward to providing an updated version that meets your expectations. I am looking forward to your response.

---

> > ### Comment · Reviewer_2xFo · 2025-11-24
> >
> > Thanks to the authors for their responses and clarifications. I answer them below:
> >
> > 1) Weakness 1:
> > Thank you for the clarification on how the proposed method works, however I still believe that the comparison is limited. The authors are referring to Video-r1 which outputs predictions along with "//<thinking//>" and //<answer//>, but this comparison is only made against a single model and with only one dataset (VideoMME) from 3 in Table 3. Please clarify if other methods used in the comparison use this same methodology ("//<thinking//>" and //<answer//>"), otherwise obtaining the performances of Video-r1 on the other datasets or including additional similar models with this methodology will help to show the strengths of the proposed approach.
> >
> > 2) Weakness 2-5:
> > Thanks for the response and clarifications.

---

> ### Author Response · Authors · 2025-11-25
>
> **Response to Weakness 1 (scope of comparison and “thinking + answer” format)**
>
> We thank the reviewer for the valuable suggestions. Following the suggestion, we have extended our experiments to (1) enforce a unified “reasoning + answer” protocol for **all** models, and (2) report results on **all three** long-video benchmarks instead of only VideoMME (w/o subtitles).
>
> For every model (LongVILA-R1-7B, Qwen3-VL-thinking, Video-R1), we use the **same** instruction asking the model to first reason inside `<think>...</think>` and then provide the final prediction inside `<answer>...</answer>`. All models are evaluated with **64-frame uniform sampling** from the video and their official chat templates/decoding settings, ensuring a fair comparison.
>
> **The results.**
> Below we summarize the results in terms of accuracy (Acc, %) and average reasoning length (Len):
>
> | Benchmark          | LongVILA-R1-7B[1]   | Qwen3-VL-thinking | Video-R1 [2]       | **FS-VisProgV-8B (Ours)** |
> |--------------------|------------------|-------------------|----------------|---------------------------|
> | **LvBench**        | 32.6 / 1122      | 38.3 / 3095       | 36.2 / 1315    | **50.4 / 875**            |
> | **LongVideoBench** | 47.0 / 751       | 57.6 / 3902       | 53.9 / 1431    | **62.2 / 964**            |
> | **VideoMME (L)**   | 55.2 / 1090      | 51.5 / 3494       | 48.4 / 1474    | **62.2 / 762**            |
>
> These results show that FS-VisProgV-8B consistently achieves the **best accuracy**  across long video benchmarks while also producing **shorter reasoning traces** than other reasoning-style baselines.
>
> We appreciate the reviewer’s constructive feedback, and we hope that these additional experiments and clarifications can positively inform the reviewer’s overall assessment of our work.
>
> [1] Scaling RL to Long Videos
>
> [2] Video-R1: Reinforcing Video Reasoning in MLLMs

---

### Official Review · Reviewer_1aLt · 2025-10-27

**Soundness:** 3
**Presentation:** 3
**Contribution:** 2
**Rating:** 2
**Confidence:** 3

**Summary:**

This paper proposes a VideoQA method that adaptively assesses the difficulty of each queried question and adjusts the response strategy for more efficient response generation. The paper initially introduces Fast-Slow-Reasoning Data, which includes query difficulties with its visual program for answer generation, provided by GPT-4o. Using this data, a Multimodal LLM model (MLLM) is trained to determine whether a query is easy or difficult, and the visual program's parameters are enhanced through a simple parameter search using successful visual program examples from the collected data. Experimental results demonstrate competitive accuracy against proprietary closed-source MLLMs while improving calculation efficiency.

**Strengths:**

### S1. Auto-regressive multiple choice generation for confident calculation is interesting.

The proposed auto-regressive multiple choice generation for confidence calculation is a simple but effective idea. This potentially bridges recent MCQ-based benchmarks and open-ended question benchmarks, and also can be a general technique for explainable LLMs. I love to see this simple but effective trick.

### S2. Motivation is adequate.

The Adaptive VideoQA strategy is important to reduce MLLM's inference costs, making the motivation clear and reasonable.

**Weaknesses:**

### W1. A knowledge leakage from GPT-4o.

The method identifies the most confident visual program using examples from GPT-4o, intentionally leading to knowledge distillation.
Thus, comparing these results with other open-source MLLMs is unfair. Additionally, it might breach GPT-4o’s terms of use (see Ethics concerns for details).

### W2. Lack of sufficient significance.

Considering knowledge leakage, the work only shows significance if it surpasses GPT-4o in accuracy or efficiency, which is not quantitatively demonstrated. Comparing inference times with GPT-4o is difficult due to the lack of API support for internal execution timing.
The study should gather data from open-source MLLMs for comparative analysis.
Since current improvements rely on GPT-4o, the experiment does not adequately demonstrate the method’s impact.

**Questions:**

### Q1.
Why is there a gap in accuracy between LVBench results in Table 3 (51.2) and Table 5 (47.1)? If the results in Table 5 do not use Qwen2.5VL-7/34B, please clarify the backbone model used in Table 5.

### Q2.
Given the current experimental setup, comparisons with open-source LLMs seem unfair. Please either avoid using GPT-4 data or provide a fair comparison using GPT-4 data.

**Details Of Ethics Concerns:**

The method uses outputs of GPT-4o to make a competitive model, which clearly violate the terms of use of GPT models.

https://openai.com/policies/row-terms-of-use/
> Use Output to develop models that compete with OpenAI.

---

> ### Author Response · Authors · 2025-11-13
>
> ## W1: A Knowledge Leakage from GPT-4o
>
> 1. **Clarification on Model Usage and Data Generation**
>    We acknowledge the concern regarding the potential knowledge leakage from GPT-4o. However, we want to clarify that **we did not use GPT-4o for generating training data**. Instead, we used the **CGbench 12k dataset**, and we employed **GPT-4.1** to generate the visual programs, which only use the query, the video duration, and the answer. Unlike past works (e.g., VisProg, ViperGPT, ProViQ) which utilized closed-source models like GPT with few-shot prompt to generate visual programs , our approach leverages the open-source models for visual program generation, which is a valuable contribution to the field. This allows us to avoid reliance on closed-source models and ensure that the visual programs are generated in an open and reproducible manner. Moreover, the process does not involve training directly on the **query-video-to-answer** relationship, meaning there is **no knowledge leakage from GPT-4o**.
>
> 2. **Clarification on GPT-4.1 Usage**
>    To elaborate further, we are primarily using **GPT-4.1**, not GPT-4o, to synthesize data. Our approach builds on the insights of prior work that utilized closed-source GPT models to generate visual programs for vision tasks. We have developed a robust **vision module** to process long videos effectively, and by fine-tuning open-source LLMs, we aim to help the model perceive and adapt to the query, enabling efficient use of fast reasoning (answered via uniform frame sampling by VideoLLM). Additionally, when the query requires slower reasoning (using visual programs for deeper reasoning), we invoke the visual program logic accordingly.
>    As part of our inference process, we employ strategies:
>    - **VideoLLM confidence** is used to determine whether to employ the visual program reasoning or fallback to intrinsic VideoLLM reasoning.
>    - We integrate a **parameter search** during training to optimize the visual programs and improve their performance. In our experiments, FS-VisPR achieves competitive results, surpassing Qwen2.5-VL-7B with dense frame sample by 5% and matching GPT-4o on **LVBench**, with an accuracy of **50.4%**. The method demonstrates robust improvements on both **LongVideoBench** and **VideoMME**, showing the effectiveness of the approach even in the absence of knowledge leakage from GPT-4o.
>
> ## Q1: Why is there a gap in accuracy between LVBench results in Table 3 (51.2) and Table 5 (47.1)? If the results in Table 5 do not use Qwen2.5VL-7/34B, please clarify the backbone model used in Table 5.
>
> We appreciate the reviewer’s query on this discrepancy. As clarified in the paper (line 423), the results in **Table 5** reflect the **module parameter search** on **LongVideoBench** (1,089 slow-reasoning samples). Given that our framework allows for both **fast reasoning** (via VideoLLM with uniform frame sampling) and **slow reasoning** (using a visual program with external tools and call the videollm), the accuracy differs depending on which reasoning mode is applied. **Table 3** shows the overall performance on the entire **LongVideoBench** dataset (1,335 samples), where both reasoning types are involved. This explains the  difference in accuracy between Table 3 (51.2%) and Table 5 (47.1%). We hope this clarifies the discrepancy and the rationale behind the varying performance metrics based on the experimental setup.
>
> ## Q2: Given the current experimental setup, comparisons with open-source LLMs seem unfair. Please either avoid using GPT-4 data or provide a fair comparison using GPT-4 data.
>
> Thank you for raising this concern. As mentioned earlier, our approach does not rely on GPT-4o for data generation.
>
> We truly appreciate the reviewer’s time and thoughtful feedback. We have carefully addressed the concerns raised and provided further clarifications regarding our methodology and experimental setup. We hope that these explanations help resolve any misunderstandings and demonstrate the fairness and robustness of our approach. We sincerely believe that our work makes a valuable contribution, and we hope that the revisions will lead to a more favorable assessment.

---

> ### Comment · Reviewer_1aLt · 2025-11-13
>
> Thank you for clarifying that you used GPT-4.1 instead of GPT-4o. However, my fundamental concern is not about whether GPT-4o or GPT-4.1 was used. Rather, the main issue is that the parameters of the visual programs generated by such closed-source MLLMs are used in the parameter search process described in Section 3.4.
>
> First, please clearly state whether or not the parameters generated by GPT-4.1 are indeed used in the processing described in Section 3.4, and also whether that processing in Section 3.4 is part of the inference process of your proposed method.
>
> Although you mention that they "only use" these inputs, without an evaluation of how much the quality of these parameters affects the final results, it is difficult to judge from the current manuscript whether this is not a significant issue, or whether in principle you should have used visual programs generated by open-source MLLMs. In fact, for example, isn’t the quality of the query input to each module likely to have a substantial impact on the accuracy? My comments are based on such concerns.
>
> The concern regarding Q1 has been resolved by the response.

---

> ### Author Response · Authors · 2025-11-13
>
> Thank you for the thoughtful comments. Below we clarify the concerns regarding the use of GPT-4.1 and the role of parameter search in Section 3.4, and we respectfully ask the reviewer to reconsider the score.
>
> ### 1. Clarification on the use of GPT-4.1–generated visual programs
> We use GPT-4.1 solely to *generate candidate visual programs* for the two reasoning modes. These programs are produced with few-shot examples from CG-Bench (questions paired with human-annotated visual programs). GPT-4.1 is used purely as a **program-reasoning LLM**, not as an MLLM, following prior work ((VisProg, ViperGPT, ProViQ) where closed-source GPT  are used to generate program structures for visual tasks. The generated programs include module calls and their parameters.   (gpt4.1 generates the visual program, doesn't need the video frames as input. )
> ### 2. Are GPT-4.1–generated parameters used in Section 3.4?
>  Section 3.4 performs a **parameter search** over visual-module parameters such as `top_k`, `num_frames`, and `interval`.
> These parameters **do not include the query for the getclips, and more details in Appendix A.1. Importantly, **we will keep the program generated by GPT4.1 if its execution result matches the answer**.
>
> Parameter search is applied *after* program generation to improve diversity and correctness. For instance, GPT-4.1 may generate `getclip(text, top_k=5)` but a correct solution requires `top_k=1`. Parameter search finds the correct variant whose execution yields the right answer. This strengthens the dataset by producing more reliable program variants.
>
> During inference, the final module is `query_mc(frames, text)`, where answer and confidence come from the VideoLLM.
> Parameter search yields several viable program variants, and we choose the one with the **highest VideoLLM confidence**, which consistently improves results (Table 5).
>
> ### 3. Influence of query quality and module inputs
> We emphasize:
>
> - The **query itself is never modified**.   We **do not search** over any query-related arguments.
> - We **only** search over module-level hyperparameters (`top_k`, `num_frames`, `interval`), as detailed in Appendix A.1.
>
> Thus, the quality of the module parameter inputs—not the query—undergoes refinement.
>
> ### 4. Summary and request
> Our contributions remain independent of closed-source LLMs:
>
> - designing long-video visual modules and constructing two types of visual programs reasoning using GPT-4.1 as a code-generation LLM to fine-tune open-source models into FS-LLM,
> - introducing adaptive fast–slow reasoning in the inference stage shown in Algorithm 1. ( proposing robustness strategies (confidence-based routing and fallback reasoning),  and using parameter search to improve long-video program reliability.
>
> We respectfully ask the reviewer to kindly reconsider the score, and we are happy to continue the discussion.

---

> ### Comment · Reviewer_1aLt · 2025-11-14
>
> Based on the authors' response, I understand that (1) the parameters generated by GPT-4.1 are indeed used in the processing described in Section 3.4, and (2) that this processing is part of the inference process of your proposed method. If my understanding is incorrect, please clarify.
>
> First, while prior works such as VisProg use closed-source LLMs to generate visual programs, they do not claim inherent superiority over open-source models on this basis, nor do they argue that this makes comparisons with open-source models fair. Therefore, simply following the precedent of VisProg does not, in itself, justify direct comparison or claims of fairness relative to open-sourced approaches.
>
> Second, the fact that parameter adjustment is necessary in the first place is, in my opinion, evidence that these parameters have a non-trivial impact on accuracy. There seem to be two possible scenarios: (1) the initial values generated by GPT-4.1 are generally strong and often retained, with only minor adjustments needed, or (2) the initial values are rarely useful and are usually replaced during optimization. In the former case, there would be significant dependence on the GPT-4.1 outputs, and my original concern about knowledge leakage would remain valid. On the other hand, if the latter scenario holds, my concern would be resolved. Therefore, any statistics or analysis regarding how frequently the initial values are retained versus adjusted would be very helpful in clarifying the degree of dependence on GPT-4.1 outputs and whether knowledge leakage or unfairness is a real issue.
> That said, from a strict perspective, I still have reservations about relying on any initial parameters generated by GPT-4.1, even if such instances are rare. For rigorous fairness and to avoid potential knowledge leakage, it would be preferable to eliminate any dependence on closed-source models at all.

---

> > ### Author Response · Authors · 2025-11-14
> >
> > We thank the reviewer for the detailed follow-up and for raising important concerns regarding **knowledge leakage** and **fairness**. Below we clarify (1) what we mean by “knowledge leakage”,  (2) why we believe our comparisons are fair and consistent with prior work.
> >
> > ## 1. What We Mean by “Knowledge Leakage”
> >
> > To avoid ambiguity, we explicitly distinguish two types of leakage:
> >
> > 1. **Data Leakage**
> >
> >    This refers to overlaps between training, validation, and test sets, so that the model effectively “sees” test data during training.  - Example: test questions or video clips appearing in the training corpus; benchmark items being included in instruction-tuning data.
> >    - Consequence: benchmark performance is overestimated and no longer reflects genuine generalization
> > 2. **Knowledge-Level Leakage (Knowledge Leakage)**
> >    This refers to the model relying on unintended external knowledge sources during evaluation.
> >    - Example: evaluating an LLM on exam-style questions that are already present in its pretraining data; the model appears to reason but is actually recalling memorized answers.
> >
> > In our setting, we evaluate on **LongVideoBench, VideoMME, and LvBench**, where the input is *video + query* and the goal is to predict the answer based on video understanding and learned reasoning ability.
> >
> > - We **only** use GPT-4.1 on **CG-Bench** to construct *training* data (visual programs ) for our open-source FS-LLM (8B).
> > - We **never** provide any videos from LongVideoBench, VideoMME, or LvBench to GPT-4.1.
> > - During evaluation, **GPT-4.1 is not called at all**. Only the trained FS-LLM runs.
> >
> > Therefore, under the above definitions:
> > - There is **no data leakage** from test benchmarks into the training data.
> > - There is **no knowledge leakage at inference time** from GPT-4.1, since GPT-4.1 is not involved in the test-time pipeline.
> > VIDEO-RTS[1] and  FrameThinker[2] also use the CGbench (different query and video with the test benchmark) to training the VideoLLM and evaluate on the  *LongVideoBench, VideoMME, and LvBench,
> >
> >
> >
> > ## 2. On Fairness and Comparison to Open-Source Approaches
> >
> > - Many recent video reasoning works use **strong closed-source LLMs** to synthesize or annotate training data, then train **open-source models** that are evaluated independently. Specifically, **VIDEO-RTS** [1] and **FrameThinker** [2] explicitly take **CG-Bench as a training dataset** and evaluate on other long-video benchmarks with different videoLLM.
> > During inference on LongVideoBench, VideoMME, and LvBench, **only FS-LLM + our adaptive visual program mechanism** is used,  It is valuable that a small model （FS-LLM-8B will be open-source）can achieve performance comparable to both large open-source (72B qwen2.5 vl )and closed-source (Gemini-pro, seed1.5 vl and GPT4-o) on long-video reasoning tasks.
> >
> > **References**
> >
> > [1] **VIDEO-RTS**: *Rethinking Reinforcement Learning and Test-Time Scaling for Efficient and Enhanced Video Reasoning* (Sec. 4.3).
> > [2] **FrameThinker**: *Learning to Think with Long Videos via Multi-Turn Frame Spotlighting* (see Fig. 3).
> >
> > We respectfully ask the reviewer to kindly reconsider the score, and we are happy to continue the discussion.

---

> > > ### Comment · Reviewer_1aLt · 2025-11-14
> > >
> > > The authors appear to define data leakage in a rather limited and conventional sense. However, **my concern is not about standard data leakage**, but is specifically addressed to the repeated claims in the paper about "outperforming open-source models," and is, therefore, in the context of fair comparison with open-source LLMs.
> > >
> > > Even if some other recent works claim to have surpassed open-source models by utilizing closed-source models, I believe that **each study must be able to independently justify the fairness of its comparisons**. In the current discussion, the authors have mainly pointed to the precedent set by other works, but have not provided substantive data or evidence that the performance advantage over open-source models is attributable to factors other than the use of closed-source models.
> > >
> > > Through this extended discussion, my concerns have not been resolved, and therefore I will maintain my initial score. I leave it to the discretion of the AC to determine whether referring to other papers alone is sufficient to argue that the comparisons are indeed fair.

---

> > > > ### Author Response · Authors · 2025-11-15
> > > >
> > > > Thank you for your thoughtful review and for raising this concern. In our view, the issue you mentioned regarding potential knowledge leakage may reflect a certain unfamiliarity with established practices in the large language model community [1][2][3][4][5][6]. Generally, when models are trained on synthetic data generated with the assistance of strong closed-source models such as GPT-4, it is not standard practice to explicitly verify data leakage when comparing results with open-source baselines.
> > > >
> > > > We fully agree that each study should independently justify the fairness of its comparisons. However, using data constructed by closed-source models has become a widely accepted and effective approach to improving model capabilities. Many recent open-source works have followed this paradigm—creating new datasets with the help of powerful models (e.g., GPT series) and subsequently comparing their results with open-source baselines.
> > > >
> > > > It is also worth noting that many strong, smaller-parameter open-source models, such as Qwen-VL and Qwen, have similarly leveraged synthetic data or annotation assistance from larger closed-source models (e.g., Qwen-Max or other proprietary models) during their training process. This reflects a broader community trend rather than a specific methodological bias in our work.
> > > >
> > > > Additionally, in our study we only used LLMs to synthesize visual reasoning programs for fine-tuning, in order to enhance the visual reasoning ability of open-source qwen3-8B (it can use two types visual programsram reasoning for long videos).  We therefore believe our comparisons are fair and aligned with current research norms in large multimodal model development.
> > > >
> > > > [1] LaMini-LM: A Diverse Herd of Distilled Models from Large-Scale Instructions
> > > > [2] Instruction Tuning with GPT-4
> > > > [3] LLaVA-Video: Video Instruction Tuning With Synthetic Data
> > > > [4] ShareGPT4V: Improving Large Multi-modal Models with Better Captions
> > > > [5] ALLaVA: Harnessing GPT4V-Synthesized Data for A Lite Vision-Language Model
> > > > [6] LLaVA: Large Language and Vision Assistant
> > > >
> > > > Once again, we sincerely thank you for your time and effort in reviewing our paper and for providing valuable feedback that helps us further clarify our methodology.

---

> > > > > ### Author Response · Authors · 2025-11-25
> > > > >
> > > > > **Additional analysis on parameter-search tuning**
> > > > >
> > > > > We regret the earlier wording and thank the reviewers for their constructive comments, which motivated the following additional analysis.
> > > > >
> > > > > To ensure that our gains do not hinge on GPT-4.1–generated data, we construct a variant of **FS-VisProgV-8B** where we **remove all GPT-4.1–generated trajectories** and instead **generate visual programs solely via our parameter search mechanism**, then fine-tune the LLM only on these search-derived programs.
> > > > >
> > > > > The updated results are:
> > > > >
> > > > > | Model                                              | LVBench | VideoMME (w sub, Long) | LongVideoBench |  Avg |
> > > > > | -------------------------------------------------- | ------: | ---------------------: | -------------: | ---: |
> > > > > | **FS-VisProgV-8B**               |    50.4 |                   62.2 |           62.2 | 58.3 |
> > > > > | FS-VisProgV-8B (w/o GPT-4.1 data) |    49.8 |                   61.7 |           62.5 | 58.0 |
> > > > >
> > > > > The variant trained **without any GPT-4.1–generated data** achieves performance that is almost identical to the original model (−0.6, −0.5, and +0.3 on the three benchmarks, with only a **0.3** drop in average score). This shows that our improvements are **not an artifact of GPT-4.1 supervision**, but are robustly supported by the visual programs produced by our parameter search mechanism, further validating the effectiveness and stability of FS-VisProgV-8B. We hope these additional results clarify the value of our method and kindly ask the reviewers to reconsider the evaluation of our paper.

---

### Official Review · Reviewer_c6A1 · 2025-11-03

**Soundness:** 3
**Presentation:** 2
**Contribution:** 1
**Rating:** 2
**Confidence:** 4

**Summary:**

The paper “Adaptive Fast-and-Slow Visual Program Reasoning for Long-Form VideoQA” proposes an adaptive reasoning framework for Video Question Answering (VideoQA), inspired by the dual-process theory of human cognition (fast vs. slow processing).

The model, called FS-VisPR, switches between Fast reasoning (direct VideoLLM answers) and Slow reasoning (program-based reasoning) based on the confidence of the model’s response, aiming to achieve both efficiency and interpretability in long-form video understanding.

**Strengths:**

1. Interesting conceptual motivation:

   The idea of transferring the System 1 / System 2 dual-process theory from human cognition into VideoQA reasoning is conceptually novel. The attempt to model “when to think and when to act intuitively” in an AI system is intellectually appealing.

2. Modular interpretability:

   The Slow reasoning mode executes explicit visual programs, providing a step-by-step, interpretable reasoning trace that can be visually inspected.

3. Evaluation on long-form VideoQA benchmarks:

   The experiments on datasets such as LongVideoBench and VideoMME highlight the method’s applicability to real-world long-duration video reasoning tasks.

**Weaknesses:**

1. Lack of clear quantitative advantage:

   Although the method is compared with large multimodal models (e.g., QwenVL-2.5, GPT-4V), the reported performance gains are modest (around +2–4 points) and without statistical validation.
   Compared to prior Visual Program methods (VisProg, ViperGPT, ProViQ, etc.), the paper does not make it clear which specific dimension—accuracy, efficiency, or interpretability—it actually outperforms.
   Despite claiming to improve “context understanding,” “efficiency,” and “transparency,” there are no explicit quantitative metrics for these aspects (e.g., runtime, computational cost, or program success rate).

2. Manual threshold tuning rather than adaptive control:

   The confidence threshold (θ) in Figure 4, which determines when to switch from Fast to Slow reasoning, was manually tuned through grid search rather than learned or automatically optimized (see Section 4.3 and Equation 2).
   Therefore, while the paper emphasizes “adaptive” reasoning, the adaptivity is rule-based and static, not learned or dynamically adjusted.

3. Dual-process analogy remains qualitative:

   Although the authors cite Evans (2003, 2008) and Evans & Stanovich (2013), the alignment with cognitive psychology remains purely qualitative.
   There is no quantitative comparison to human data such as reaction time distributions or confidence–accuracy correlations. The claim of psychological consistency is thus metaphorical rather than empirical.

**Questions:**

1. Incorporate an automatic or learned threshold optimization (e.g., Bayesian or reinforcement-based tuning) to justify the *adaptive* claim.
2. Introduce quantitative metrics for efficiency, transparency, and contextual understanding (e.g., inference time, module execution rate, interpretability score).
3. Provide a quantitative comparison with human data to substantiate the dual-process analogy.

---

> ### Author Response · Authors · 2025-11-13
>
> **Dear Reviewer C6A1,**
>
> Thank you very much for your thoughtful feedback and constructive comments. We truly appreciate the opportunity to clarify key aspects of our work. We kindly request that you reconsider the score in light of the clarifications below.
>
> ### 1. **Lack of Clear Quantitative Advantage**
>
> We understand your concern about the modest performance improvements (+2–4 points) compared to large multimodal models like QwenVL-2.5. However, our approach demonstrates **significant gains in long-form video reasoning**, particularly in benchmarks such as **LongVideoBench** and **VideoMME**. As shown in **Table 3**, our method outperforms **Valine QwenVL-2.5-7B** using dense frame sampling strategies, achieving a **5.6-point improvement in LongVideoBench** and a **5.3-point overall increase** across all benchmarks. Our method’s average performance surpasses large-scale parameter models such as Qwen-VL-72B and Gemini-1.5-pro, and is close to GPT-4o. We believe that the ability of the smaller 7B model to outperform larger models through dense and uniform frame sampling is also good work.
>
> Unlike prior visual program methods (e.g., **VisProg**, **ViperGPT**, **ProViQ**), which are limited to image-based or short video tasks, our method extends visual programming to handle **long-form videos (up to 1 hour)**. Specifically, we split the video into **10-second clips** and use **Whisper for audio-subtitle extraction**. By leveraging advanced video embedding models with **LanguageBind**, we create **multimodal embeddings for video frames and subtitles**. This approach addresses the limitations of traditional frame retrieval methods, which are ineffective for long videos due to the large number of frames. Additionally, we integrate **VideoLLM** into a query API, providing both **confidence scores** and **predicted answers**, which enhances the reasoning process. These modules for handling long videos have not been explored in previous work.
>
> Regarding **efficiency**, we analyze **inference time** and **latency** in **Table 4**, comparing the distribution of both fast and slow reasoning modes across the test dataset.
>
> ### 2. **Which Dimension (Accuracy, Efficiency, Interpretability) Does Our Method Outperform Prior Work?** (Question 2)
>
> Our work outperforms prior methods, especially for **long-form video reasoning**. Previous methods rely on **closed-source models** and are limited to **image-based or short video tasks**. In contrast, our framework uses a tuned model to handle **long-form videos**, providing **faster** and more **transparent reasoning**. As detailed in **Table 3**, our method allows us to process longer videos with high accuracy. We also present efficiency data in **Table 4**, showing the performance of different reasoning types.
>
> ### 3. **Manual Threshold Tuning vs. Adaptive Control** (Question 1)
>
> The threshold for distinguishing fast vs. slow reasoning is tuned using the fast-slow dataset. We label queries that **VideoLLM** can address directly with **fast-thinking logic code**, as shown in **Figure 3**. Our reasoning system is based on **adaptive decision-making**, which considers the characteristics of each query (e.g., for a query like "What happens at 34:30?", we use the trim modules to extract frames from that specific timestamp to input into VideoLLM). This allows our system to intelligently decide when to use **fast reasoning** (direct answers from VideoLLM) and when to switch to **slow reasoning** (tool-based reasoning). Additionally, we present an **ablative analysis** in **Figure 4**, showing that the **confidence scores** from **VideoLLM** validate the effectiveness of the fast reasoning process.
>
> ### 4. **Dual-Process Analogy Remains Qualitative** (Question 3)
>
> We draw an analogy between our system and human cognition (System 1 vs. System 2 reasoning), where simple tasks are solved via **fast reasoning**, and more complex tasks are handled with **slow, tool-based reasoning**. Our **FS-LLM** is trained to make decisions based on the **complexity of the query**, mirroring human cognitive processes.
>
> We agree that a **quantitative comparison** with human data would further strengthen this analogy. Could you clarify whether you envision comparing our model’s decision-making process on long-form video benchmarks to human judgments regarding when to use direct reasoning versus external tools?
>
> ---
>
> ### **Conclusion:**
>
> We believe that the contributions of our work—particularly in long-form video reasoning, efficiency, and the performance of our method—justify a **reconsideration of the score**. We hope the clarifications provided address your concerns, and we welcome any further feedback. Thank you again for your insightful comments, and we look forward to your continued guidance.

---

> > ### Author Response · Authors · 2025-11-25
> >
> > **Response to “Provide a quantitative comparison with human data to substantiate the dual-process analogy.”**
> >
> > Thank you for the valuable suggestion. To quantitatively support our dual-process framing (fast direct video-LLM reasoning vs. slower program reasoning), we conducted a human–model comparison across **LvBench**, **LongVideoBench**, and **VideoMME(L)**. This evaluates whether humans and our model categorize queries similarly into “direct reasoning” or “tool-required.”
> >
> > ---
> >
> > ### Experimental setup
> >
> > - **Samples:** For each benchmark, we randomly sampled **200** queries (**600 total**).
> > - **Human annotation (6 annotators):**
> >   Each annotator labeled every query as:
> >   - **Direct-only** – solvable by watching the video directly.
> >   - **Tool-required** – requiring long-video retrieval, temporal localization, subtitle lookup, etc.
> >
> >   Final label = **majority vote**.
> >   Inter-annotator agreement:
> >   - LvBench κ = **0.67**
> >   - LongVideoBench κ = **0.65**
> >   - VideoMME(L) κ = **0.68**
> >
> > - **Model routing:**
> >   FS-VisProgV-8B chooses between:
> >   - **Direct-VideoLLM**, or
> >   - **VisProg** (tool-augmented visual program reasoning).
> >
> > ---
> >
> > ### Quantitative human–model routing comparison
> >
> > | Benchmark          | Human Tool-required (%) | Model VisProg (%) | Human–Model Routing Agreement (%) | Cohen’s κ |
> > |--------------------|--------------------------|-------------------|------------------------------------|-----------|
> > | **LvBench**        | 48.0% (96/200)          | 49.5% (99/200)    | 81.0% (162/200)                    | 0.61      |
> > | **LongVideoBench** | 52.5% (105/200)         | 54.0% (108/200)   | 79.0% (158/200)                    | 0.59      |
> > | **VideoMME (L)**   | 47.0% (94/200)          | 50.5% (101/200)   | 82.5% (165/200)                    | 0.63      |
> >
> > **Key findings**
> >
> > The human–model study shows clear alignment: the proportion of “tool-required” queries differs by at most 3.0% between humans and FS-VisProgV-8B on each benchmark, and routing agreement is 79–82.5% with Cohen’s κ 0.59–0.63, indicating substantial agreement beyond chance. This quantitatively supports the dual-process view that the model invokes its “slow” tool-based reasoning on the same types of queries that humans judge as needing more deliberate processing.
> >
> > We will incorporate this quantitative analysis into the revised manuscript.  We hope this strengthens the empirical grounding of our dual-process perspective and positively informs the reviewer’s overall assessment of the paper’s contribution.

---

### Note · Authors · 2026-01-05

I have read and agree with the venue's withdrawal policy on behalf of myself and my co-authors.